# DIAPH3 deficiency links microtubules to mitotic errors, defective neurogenesis, and brain dysfunction

Eva On-Chai Lau[1‡], Devid Damiani[1†§], Georges Chehade[1†], Nuria Ruiz-Reig[1], Rana Saade[1], Yves Jossin[2], Mohamed Aittaleb[3], Olivier Schakman[4], Nicolas Tajeddine[4], Philippe Gailly[4], Fadel Tissir[1,3]*

[1]Université catholique de Louvain, Institute of Neuroscience, Developmental Neurobiology, Brussels, Belgium; [2]Université catholique de Louvain, Institute of Neuroscience, Mammalian Development and Cell Biology, Brussels, Belgium; [3]College of Health and Life Sciences, HBKU, Doha, Qatar; [4]Université catholique de Louvain, Institute of Neuroscience, Cell Physiology, Brussels, Belgium

*For correspondence:
fadel.tissir@uclouvain.be

[†]These authors contributed equally to this work

Present address: [‡]Division of Life Science, The Hong Kong University of Science and Technology, Hong Kong, China; [§]Center for RNA Technologies, RNA Lab, Center for Human Technologies, IstitutoItaliano di Tecnologia, Genova, Italy

**Abstract** Diaphanous (DIAPH) three (DIAPH3) is a member of the formin proteins that have the capacity to nucleate and elongate actin filaments and, therefore, to remodel the cytoskeleton. DIAPH3 is essential for cytokinesis as its dysfunction impairs the contractile ring and produces multinucleated cells. Here, we report that DIAPH3 localizes at the centrosome during mitosis and regulates the assembly and bipolarity of the mitotic spindle. DIAPH3-deficient cells display disorganized cytoskeleton and multipolar spindles. DIAPH3 deficiency disrupts the expression and/or stability of several proteins including the kinetochore-associated protein SPAG5. DIAPH3 and SPAG5 have similar expression patterns in the developing brain and overlapping subcellular localization during mitosis. Knockdown of SPAG5 phenocopies DIAPH3 deficiency, whereas its overexpression rescues the DIAHP3 knockdown phenotype. Conditional inactivation of *Diaph3* in mouse cerebral cortex profoundly disrupts neurogenesis, depleting cortical progenitors and neurons, leading to cortical malformation and autistic-like behavior. Our data uncover the uncharacterized functions of DIAPH3 and provide evidence that this protein belongs to a molecular toolbox that links microtubule dynamics during mitosis to aneuploidy, cell death, fate determination defects, and cortical malformation.

## Introduction

Development of the cerebral cortex requires the production and positioning of the right number of neurons. At initial stages of cortical development, the dorsal telencephalon is organized in a pseudostratified epithelium consisting of neural stem cells (NSCs, also known as neuroepithelial cells) that undergo multiple rounds of proliferative division to expand the initial pool of progenitors. Once neurogenesis begins, the neocortex comprises two germinal zones: the ventricular zone (VZ), which forms the lining of lateral ventricles and contains radial glial cells (RG), also known as apical neural progenitor cells (aNPCs), and the adjacent subventricular zone (SVZ), which is located dorsally to the VZ and contains basal progenitors (BP). In the VZ, aNPCs undergo several rounds of divisions to self-renew and generate glutamatergic neurons. aNPCs can also give rise to BP cells, which delaminate from the VZ and translocate to the SVZ, where they divide a limited number of times to increase the final output of neurons (*Florio and Huttner, 2014*; *Noctor et al., 2008*). A delicate balance between proliferation and differentiation of aNPC must be preserved during neurogenesis. This balance is regulated by intrinsic and extrinsic factors, and involves rearrangements of the cytoskeleton to support a rigorous sequence of fate decisions. During cell division, filamentous actin rearranges at the

cell cortex to enhance cell membrane rigidity for the anchorage of astral microtubule (MT) (*Heng and Koh, 2010*). In the meantime, the centrosome, also known as the microtubule organising centre (MTOC), duplicates, and the two nascent centrosomes migrate toward the poles of the cell. Astral and spindle MT nucleate from the centrosomes and extend to the cortex and equator of the cell, respectively. Polarity proteins G-protein signaling modulator 2 (GPSM2, aka PINS/LGN) and nuclear mitotic apparatus (NUMA) are distributed underneath the cell cortex and interact with cortical actin to connect microtubule plus-end motor proteins dynein/dynactin, and pull on astral MT (*Morin and Bellaïche, 2011*). On the other hand, spindle MT grow inwardly and attach to chromosomes at the metaphase plate. Once the chromosomes are properly aligned and each chromosome is bilaterally connected to two spindle MT, cohesin is degraded and sister chromatids migrate to opposing poles of the cell, thus enabling nuclear division (aka karyokinesis). Thereafter, actin redistributes to the contractile ring, which constricts, creating the cleavage furrow. The cell cycle is completed by cytokinesis that splits the cytoplasm. Hence, the coordinated action of actin and MT is key to cell division. Errors in centrosome duplication, actin or microtubule polymerization, spindle assembly, or chromosome segregation lead to aneuploidy and/or mitotic catastrophe.

Formins are key regulators of actin dynamics. There are fifteen mammalian formins and they all possess two formin homolog (FH) domains (*Breitsprecher and Goode, 2013*). FH1 delivers profilin-bound actin monomers to the actin filament-barbed ends accelerating elongation, whereas the FH2 dimerizes, enabling the bundling of actin filaments (*Kovar, 2006*). Diaphanous (DIAPH) formins form a subgroup of three members that have three regulatory domains: a GTPase-binding domain (GBD) at the N terminus, a diaphanous autoregulatory domain (DAD), and a diaphanous inhibitory domain (DID). DAD binds to DID, maintaining the DIAPH proteins in an inactive state. Activation occurs through the binding of Rho-GTPases, which releases the DAD/DID interaction. The best-characterized functions of diaphanous three (DIAPH3, also referred to as mDia2) are actin-related. In proliferating cells, DIAPH3 is required for the formation of the contractile ring and the cleavage furrow to enable cytokinesis (*Chen et al., 2017*; *DeWard and Alberts, 2009*; *Watanabe et al., 2013*). DIAPH3 is also involved in filopodia assembly, supporting cell migration (*Stastna et al., 2012*), and mesenchymal-amoeboid transition (*Hager et al., 2012*; *Morley et al., 2015*). In addition to the abovementioned roles, evidence for DIAPH3 implication in actin/cytokinesis-independent functions has emerged. Early in vitro studies have suggested that DIAPH3 interacts with EB1 and APC, which localizes to plus-end, thereby stabilizing MT (*Wen et al., 2004*; *Bartolini et al., 2008*). Analysis of the full *Diaph3* knockout mice has revealed an important role of DIAPH3 in the biology of NSCs. The lack of DIAPH3 severely compromises chromosome segregation, leading to aneuploidy, mitotic catastrophe, and loss of these cells (*Damiani et al., 2016*). Yet, the precise function of DIAPH3 in nuclear division, especially in cytoskeletal rearrangements, remains elusive. Furthermore, the impact of DIAPH3 deficiency on brain development and function was not assessed due to early embryonic lethality.

Here, we report that DIAPH3 localizes at the centrosome during mitosis and regulates the assembly of MT as well as the bipolar shape and orientation of the mitotic spindles. DIAPH3 deficiency disrupts the expression and/or stability of several proteins, leading to multipolar spindles and disorganized cytoskeleton. One of the affected proteins is SPAG5 (also known as the mitotic spindle-associated protein 126 or Astrin), which localizes at the centrosome and kinetochore, and has a documented function in cell division. SPAG5 displays a similar expression pattern as DIAPH3 in the developing cortex. Downregulation of SPAG5 phenocopies DIAPH3 deficiency, whereas its overexpression rescues the DIAPH3 knockdown phenotype. We also used a conditional approach to delete *Diaph3* specifically in the cerebral cortex and report that this causes a marked depletion of cortical neurons, microcephaly, locomotor impairment, and social interaction defects.

## Results

### DIAPH3 localizes to centrosome and is required for assembly and function of mitotic spindle

We studied the subcellular distribution of DIAPH3 by immunofluorescence in U2OS cells. The protein co-localized with the centrosomal marker γ-tubulin during the whole mitosis (*Figure 1A–D*). In telophase, it was also seen at the midbody (*Figure 1D*), which is consistent with its documented role in

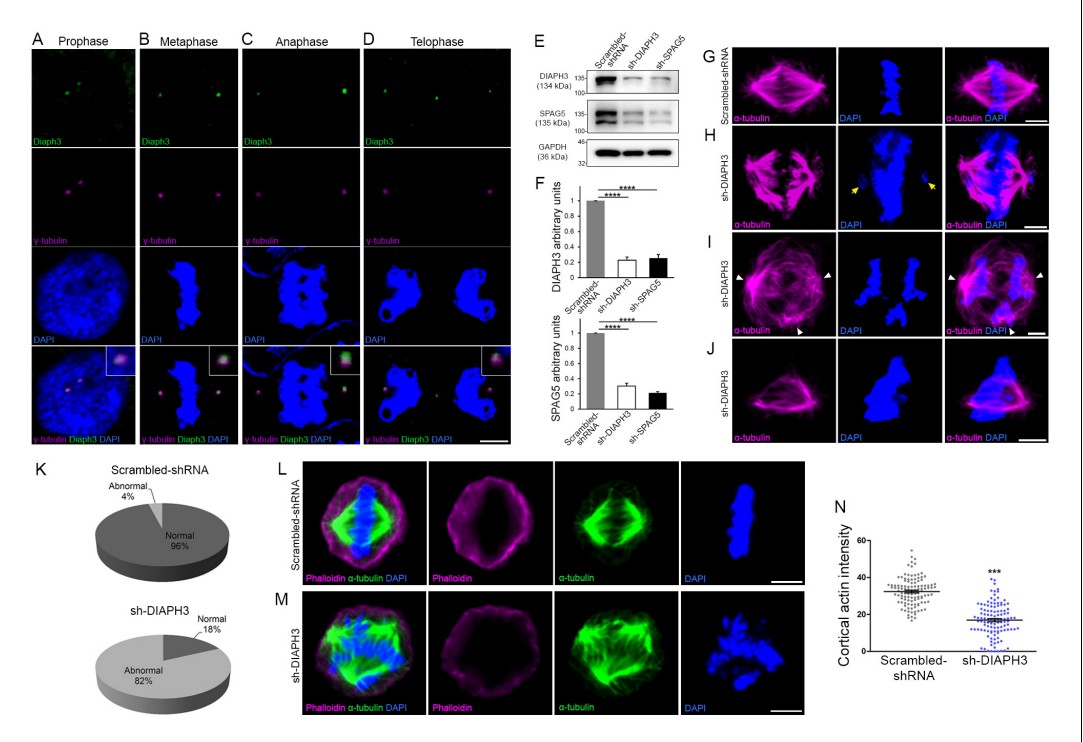

**Figure 1.** Loss of DIAPH3 disrupts the expression/stability of SPAG5 and causes mitotic defects. (A–D) Immunostaining of U2OS cells at prophase (A), metaphase (B), anaphase (C), and telophase (D), with anti-DIAPH3 (green) and anti-γ tubulin (magenta). The chromosomes were counterstained with DAPI (blue). Diaphanous three (DIAPH3) localizes at the centrosome at all mitotic stages. Insets are zooms in the centrosomal region, showing that the DIAPH3 signal is pericentrosomal. Scale bar, 10 μm. $n$ = 20 cells for each phase from three distinct experiments. (E, F) Western blot analysis of DIAPH3 and SPAG5 levels upon shRNA downregulation in U2OS cells. shRNA against DIAPH3 (sh-DIAPH3) significantly reduced the levels of DIAPH3 and SPAG5 proteins (p=0.000044 and 0.000046, respectively). Reciprocally, shRNA against SPAG5 (sh-SPAG5) reduced SPAG5 and DIAPH3 protein levels (p=0.000001 and 0.000097, respectively). Scrambled shRNA were used as control for the transfection and GAPDH as loading control. $n$ = 3 independent experiments, Student's $t$-test; error bars represent s.e.m. (G–J) Representative images of mitotic cells transfected with scrambled shRNA (G) and shRNA against DIAPH3 (sh-DIAPH3) (H–J), immunostained with anti-α tubulin (magenta). Chromosomes were counterstained with DAPI (blue). Yellow arrows in (H) point to lagging chromosomes. Arrowheads in (I) depict poles of mitotic spindle. (J) Representative image of a mitotic cell with disorganized/shrunk microtubule (MT), single centrosome/pole, and asymmetric metaphase plate. Scale bar, 5 μm. (K) Quantification of mitotic errors. 82% of DIAPH3 knockdown cells exhibit mitotic abnormalities (4% in control cells). $n$ = 118 and 115 cells from five individual experiments of scrambled shRNA and sh-DIAPH3, respectively. (L–N) Diminished cortical actin in DIAPH3 knockdown cells. (L, M) Illustration of mitotic cells transfected with scrambled shRNA (L) or sh-DIAPH3 (M), and immunostained with phalloidin (magenta) and α-tubulin (green). Chromosomes were counterstained with DAPI; scale bar, 5 μm. (N) Quantification of cortical actin (fluorescence intensity) showing a reduction of 48% of cortical actin in DIAPH3 knockdown cells. $n$ = 113 cells per condition from three distinct experiments. Student's $t$-test, p=1.07 × 10$^{-32}$. Error bars represent s.e.m.

The online version of this article includes the following source data and figure supplement(s) for figure 1:

**Source data 1.** Downregulation of DIAPH3 and SPAG5.

**Figure supplement 1.** DIAPH3 knockdown reduced cell survival and overexpression of SPAG5 rescued the phenotype.

**Figure supplement 1—source data 1.** Downregulation of DIAPH 3 and SPAG5 in U2OS cells.

the ingression of the cleavage furrow during cytokinesis (*Watanabe et al., 2013*; *DeWard and Alberts, 2009*). We downregulated the expression of DIAPH3 by shRNA (*Figure 1E,F*) and found that this disrupted cell division (82% of DIAPH3 knockdown cells showed abnormal division versus 4% in control cells; *Figure 1G–K*, *Supplementary file 1*). Notably, DIAPH3 knockdown compromised the integrity of the centrosome and bipolar shape of the spindle (40% of cells exhibited an abnormal number of centrosomes), as well as the organization of MT, leading to mis-segregation of chromosomes (*Supplementary file 1*). Knockdown of DIAPH3 also reduced cell survival to 59.1 ± 0.73% of control cells (*Figure 1—figure supplement 1A*), likely by inducing apoptosis (percentage of aCas3$^+$ cells: 1.1 ± 0.12% for scrambled shRNA versus 3.2 ± 0.16% for Diaph3 shRNA; *Figure 1—figure supplement 1B*). However, it did not affect cell proliferation as the ratio of Ki67$^+$ cells was preserved (*Figure 1—figure supplement 1C*).

## Downregulation of DIAPH3 alters SPAG5 expression

Spindle positioning during mitosis is governed by multiple mechanisms. Key among those is the 'cortical pulling', which refers to the capacity of specific sites on the cell cortex to capture and exert forces on astral MT to orient the mitotic spindle. Cortical pulling is steered by cytocortical proteins, which involve NUMA and GPSM2 (also known as LGN or partner of inscuteable [PINS]); polarity proteins PAR3 and NUMB; and the adapter protein Inscuteable (INSC). The assembly of NUMA-GPSM2 beneath cortical actin recruits the dynein/dynactin motor protein complex to haul astral MT (*Morin and Bellaïche, 2011*). Downregulation of DIAPH3 dramatically reduced the amount of cortical actin (*Figure 1L–N*). We used Western blotting on dorsal telencephalon lysates from cortex-specific *Diaph3* knockout mice (hereafter referred to as *Diaph3* cKO, obtained by crossing a floxed allele with mice expressing the recombinase Cre under the control of *Emx1* promoter; *Figure 2—figure supplement 1*) to assess the level of core cortical proteins NUMA, GPSM2, PAR3, NUMB, INSC, dynein, and dynactin; MT plus-end-associated proteins sperm-associated antigen (SPAG 5) and Kinastrin (KNSTRN, aka small kinetochore-associated protein/SKAP); and the cytoplasmic linker-associated protein (CLASP)1. We also used CENPA, a protein downregulated in DIAPH3-depleted cells (*Liu and Mao, 2016*) as the positive control for Western blotting and normalized the results to GAPDH levels. We did not detect significant changes in the level of endogenous NUMA, dynein, dynactin, INSC, or CLASP1 between *Diaph3* cKO and control mice. In contrast, polarity proteins GPSM2, NUMB, and PAR3, and microtubule-associated proteins SPAG5 and KNSTRN were downregulated (*Figure 2A,B*, *Supplementary file 2*). We studied the subcellular distribution of differentially expressed proteins in *Diaph3* cKO (GPSM2 and PAR3) or knockdown cells (NUMB, SPAG5, and KNSTRN). All displayed mild to important alterations of their partition during mitosis (*Figure 2C–H*, *Figure 2—figure supplement 2*). The distribution of GPSM2, NUMB, SPAG5, and KNSTRN in the pericentrosomal region was compromised as was that of PAR3 in the apical surface, even though we cannot exclude that some of these changes are due to the reduction in protein levels. The most evident change was observed in SPAG5. In control cells, the protein localized to the spindle pole and kinetochore during metaphase and progressively concentrated in the pericentrosomal region as mitosis proceeded to anaphase (*Figure 2C,D*). In DIAPH3-depleted cells, SPAG5 expression declined sharply (*Figure 1E,F*; *Figure 2C–H*), and its distribution was disrupted (*Figure 2E–H*, *Supplementary file 1*). The bipolar organization of the spindle was altered, and astral MT were shrunk. The chromosomes failed to congress at the metaphase plate and did not migrate to the cell poles at anaphase (*Figure 2E–H*). Importantly, the expression pattern of the *Spag5* transcript resembled that of *Diaph3* with a strong expression in cortical progenitor cells (http://www.eur-express.org/ee/databases/assay.jsp?assayID=euxassay_004999&image=01; *Damiani et al., 2016*). To explore further the relationship between DIAPH3 and SPAG5, we conducted real-time RT-PCR (*Figure 2I*) and RNAscope in situ hybridization (*Figure 2J*), both of which showed a reduction of Spag5 mRNA in *Diaph3* cKO.

## Downregulation of SPAG5 phenocopies the DIAPH3 phenotype

SPAG5 is an essential component of the mitotic spindle. It is required for chromosome alignment, sister chromatid segregation, and progression to anaphase (*Mack and Compton, 2001*; *Gruber et al., 2002*; *Thein et al., 2007*). SPAG5 promotes microtubule-kinetochore attachments and regulates the localization of several centrosomal proteins (e.g. CDK2, CDK5RAP2, CEP152, WDR62, and CEP63; *Kodani et al., 2015*). Silencing SPAG5 in U2OS cells triggered multipolar spindles and chromosome mis-segregation, a phenotype reminiscent of DIAPH3 knockdown (*Figure 1G–J* and *Figure 3A–D*; *Thein et al., 2007*). It also compromised the expression and/or stability of DIAPH3 (*Figure 1E,F* and *Figure 3A–D*). Remarkably, overexpression of SPAG5 significantly rescued the DIAPH3 knockdown phenotype and partially restored cell viability (survival: 87.1 ± 1.71% in SPAG5 rescue versus 59.1 ± 0.73% in DIAPH3 knockdown; aCas3$^+$ cells: 1.1 ± 0.15% in SPAG5 rescue versus 3.2 ± 0.16% in DIAPH3 knockdown; *Figure 1—figure supplement 1A,B*) and spindle bipolarity (*Figure 1—figure supplement 1D*). These results, along with the expression of SPAG5 and its downregulation in the *Diaph3* cKO mice, incited us to investigate its function in vivo. After testing the efficiency of SPAG5 shRNA in NIH3T3 cells (*Figure 3E,F*), we electroporated e13.5 embryos in utero with scrambled shRNA or sh-SPAG5 cDNA constructs and analyzed cell division of aNPC at e15.5. We used γ-tubulin (centrosome) and DAPI (chromosomes) to envision the orientation

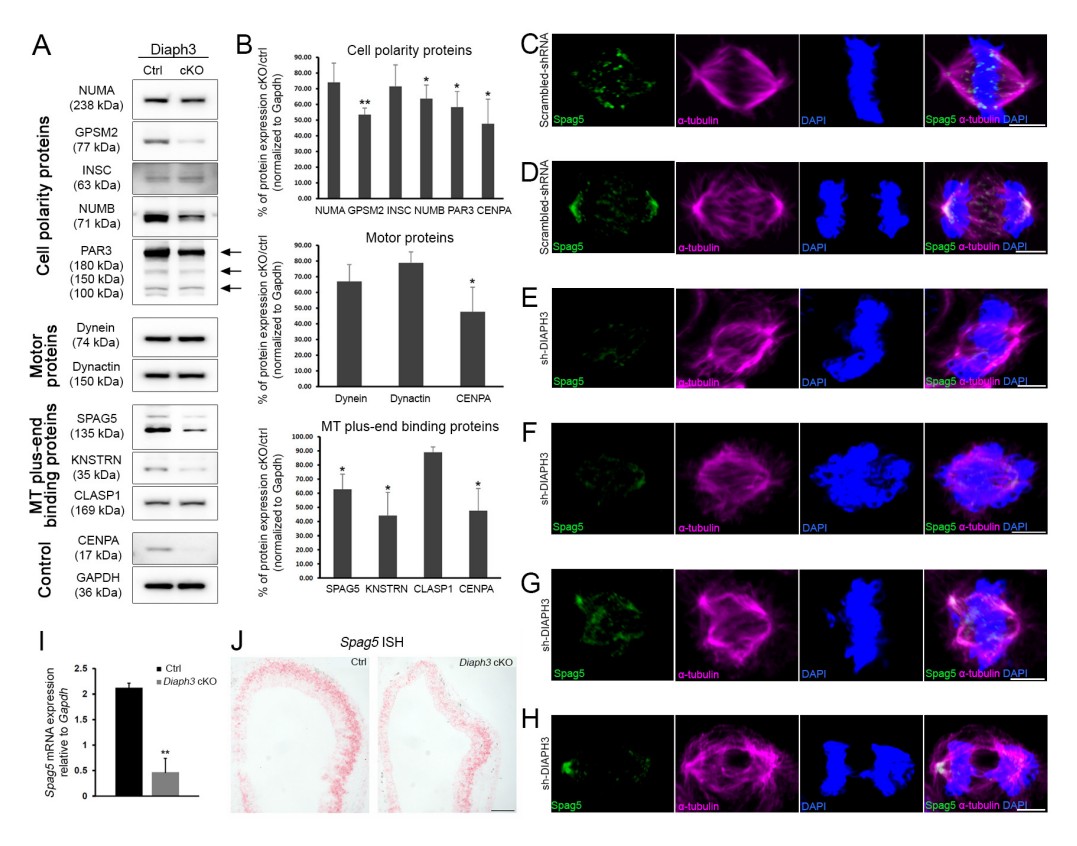

**Figure 2.** Diaph3 deficiency impairs the expression and stability of mitotic spindle polarity proteins. (A, B) Assessment of nuclear mitotic apparatus (NUMA), G-protein signaling modulator 2 (GPSM2), INSC, NUMB, PAR3, Dynein, Dynactin, SPAG5, KNSTRN, and CLASP1 levels in telencephalon extracts of *Diaph3* cKO embryos by western blotting. CENPA was used as positive control (**Liu and Mao, 2016**) and GAPDH as loading control for quantification. Both the higher and lower bands of SPAG5 were quantified. n = 4 embryos for each genotype. *p<0.05, ** p<0.01, Student's *t*-test; error bars represent s.e.m. Fold change and p-value are listed in **Supplementary file 2**. (C–H) Representative images of mitotic cells transfected with scrambled shRNA (C, D) or DIAPH3 shRNA (sh-DIAPH3, E–H), and immunostained with anti-SPAG5 (green) and anti-α tubulin (magenta) antibodies. Chromosomes were counterstained with DAPI (blue). SPAG5 localized at the centrosome and kinetochore in control cells (C, D), and both its expression level and distribution were altered in DIAPH3 knockdown cells (E–H). Scale bar, 5 μm. n = 118 and 115 cells from five individual experiments of scrambled shRNA and sh-DIAPH3, respectively. (I) Quantification of the *Spag5* mRNA by real-time RT-PCR. There is a reduction of 78% in *Diaph3* cKO relative to control mice (p=0.0046). n = 3 from nine embryos for each genotype. (J) Coronal sections of e11.5 forebrain from control (left) and *Diaph3* cKO (right, hybridized with a *Spag5* fast red-labelled RNAscope probe). The expression of *Spag5* mRNA is downregulated in *Diaph3* cKO; scale bar, 150 μm. n = 3 for each genotype.

The online version of this article includes the following source data and figure supplement(s) for figure 2:

**Source data 1.** Quantification of SPAG5 mRNA in DIAPH3 cKO cortex.

**Figure supplement 1.** Generation of cortex-specific knockout (*Diaph3* cKO) mice.

**Figure supplement 2.** KO/KD of DIAPH3 disrupts the localization of cytocortical proteins.

of the spindle at anaphase, and assess the mode of cell division (**Figure 3G–M**). We observed multi-polar spindles and a bias toward non-planar divisions in brains electroporated with sh-SPAG5 (**Figure 3H–K**). 49% (67/137) of SPAG5 knockdown cells underwent non-planar divisions versus 31% (45/147) in control cells (**Figure 3L,M**).

## Cortex-specific inactivation of *Diaph3* disrupts neurogenesis

The lack of DIAPH3 compromises nuclear division in NSCs and causes apoptosis (**Damiani et al., 2016**). To assess the effect of DIAPH3 loss on cortical neurogenesis, we performed aCas3 immunostaining at E11.5 and found that around 18% of NSCs were aCas3 positive in *Diaph3* cKO mice (18 ± 0.83% of cells in *Diaph3* cKO versus 2.4 ± 0.65% in control; **Figure 4—figure supplement 1**). We quantified the number of aNPCs (Pax6), BP (Tbr2), and neurons (Tbr1) at E13.5, and found that the

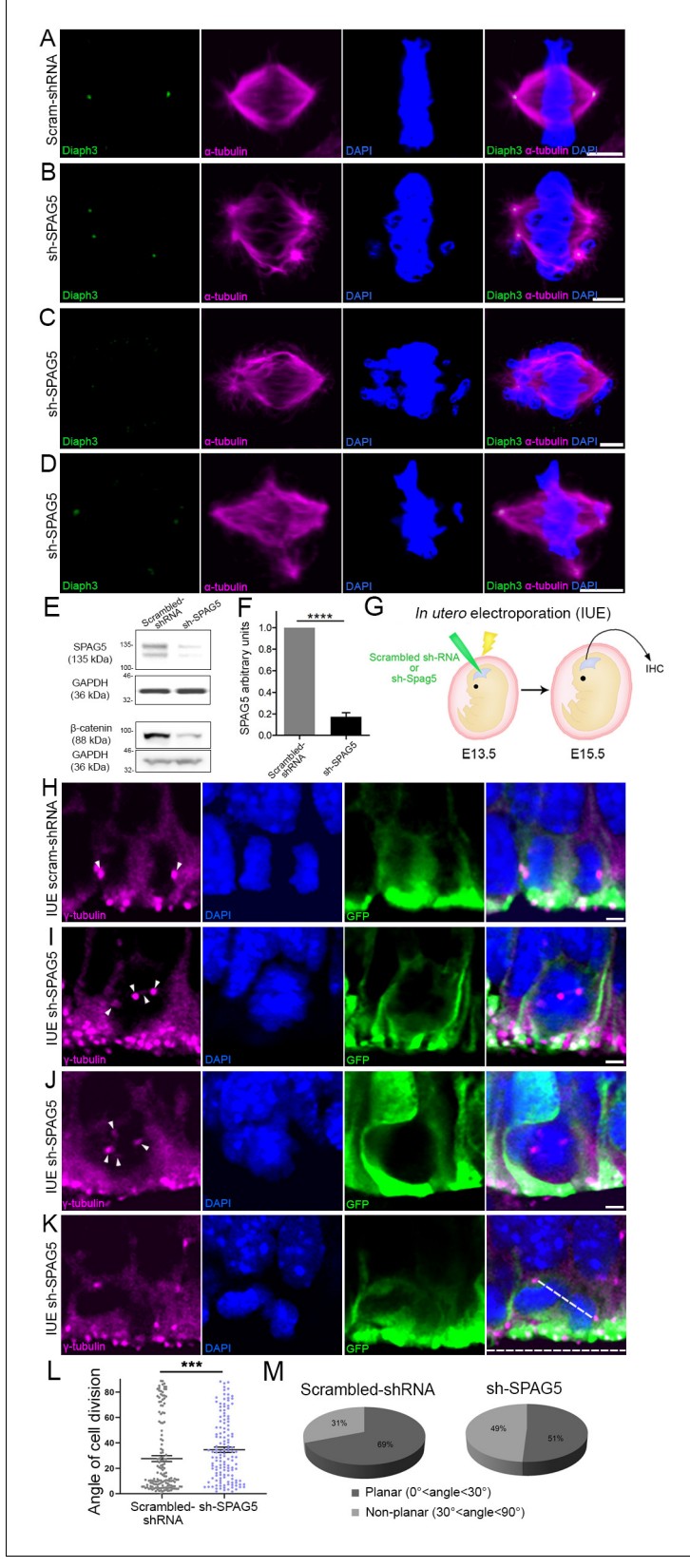

**Figure 3.** Downregulation of SPAG5 phenocopies DIAPH3 deficiency. (A–D) Representative images of mitotic cells transfected with scrambled shRNA (A) or SPAG5 shRNA (sh-SPAG5) (B–D) and immunostained with anti-DIAPH3 (green) and anti-α tubulin (magenta) antibodies. Chromosomes were counterstained with DAPI (blue). SPAG5

*Figure 3 continued on next page*

*Figure 3 continued*

deficiency downregulated diaphanous three (DIAPH3) and caused mitotic errors. Note the mis-localization of DIAPH3 and spindle abnormalities in sh-SPAG5-transfected cells. Scale bar, 5 μm. *n* = 100 cells per condition from five distinct experiments. (E, F) Western blot analysis of SPAG5 levels upon shRNA-mediated downregulation in NIH3T3 cells. Transfection of shRNA against SPAG5 (sh-SPAG5) significantly reduced the protein levels (−82.8%, p=0.00001, *n* = 3 independent experiments, Student's *t*-test; error bars represent s.e.m.). Beta-catenin, a protein downregulated by knockdown of SPAG5 (*Liu et al., 2019*), was used as control of knockdown efficiency. (G) Schematic representation of in utero electroporation of sh-SPAG5 in cortical progenitors at e13.5 and immunohistochemistry (IHC) analysis at e15.5. (H–K) Ventricular zone of telencephalic sections of e15.5 embryos electroporated in utero with scrambled shRNA and GFP (H) or SPAG5 shRNA and GFP (I–K) at e13.5, and immunostained with anti-γ tubulin (magenta) at e15.5. White arrowheads depict normal centrosomes in (H) and numerical abnormalities of the centrosome in (I–K). (K) Illustration of a neural progenitor undergoing a non-planar division. The horizontal and oblique dashed lines delineate the ventricular surface and mitotic spindle orientation, respectively. When the mitotic spindle is parallel to the ventricular surface (0°≤angle<30°), the cell division is planar. When it is oblique (30°≤angle<60°) or perpendicular (60°<angle<90°) to the ventricular surface, the division is non-planar. Scale bar = 2 μm. *n* = 5 embryos for both scrambled shRNA and sh-SPAG5. (L, M) Quantification of cell division modalities in electroporated apical neural progenitor cells (aNPCs). There is a bias toward non-planar divisions (larger angles) in sh-SPAG5-electroporated aNPCs when compared with scrambled shRNA, p=0.0009. Student's *t*-test; error bars represent s.e.m. (L). 49% (67/137 cells) of sh-SPAG5 progenitors undergo non-planar division compared to 31% (45/147 cells) in scrambled shRNA (M).

The online version of this article includes the following source data for figure 3:

**Source data 1.** Quantification of cell division modalities.

---

three populations were affected (*Figure 4A–I*). Their respective numbers decreased by 36% (number of Pax6+ cells/0.1mm$^2$ = 502 ± 48 in *Diaph3* cKO versus 786 ± 35 in control; *Figure 4A–C*), 31% (number of Tbr2+ cells/0.1mm$^2$ = 234 ± 4.8 in *Diaph3* cKO versus 339 ± 15 in control; *Figure 4D–F*), and 23% (number of Tbr1+ cells/0.1mm$^2$ = 404 ± 12 in *Diaph3* cKO versus 524 ± 18 in controls; *Figure 4G–I*). These results show that the loss of DIAPH3 has an effect on the three cortical cell types, with aNPC being the most affected population. BP and neurons are produced by neurogenic division and settle at more basal positions than their precursors. This type of division supposedly involves asymmetric inheritance of fate determinants, and has been associated with a mitotic spindle that is oblique or perpendicular to the ventricular surface, even though this view has been disputed (*Chenn and McConnell, 1995*; *Shitamukai et al., 2011*; *Shitamukai and Matsuzaki, 2012*; *Noctor et al., 2008*). During asymmetric divisions, cellular components and/or molecules regulating fate decision (e.g. the apical process, mother vs daughter centriole, cilia remnants, or polarity proteins) are believed to be inherited unevenly by the daughter cells. Given the deregulation of polarity proteins NUMB and PAR3 upon DIAPH3 downregulation, we analyzed the mitotic spindle orientation in dividing aNPCs in vivo (*Figure 4J*). We considered dividing cells in both VZ and SVZ. In control embryos, 87% (77/89) of divisions were planar and 13% (12/89) were non-planar. In *Diaph3* cKO, the fraction of planar divisions dropped to 46% (37/81), whereas that of non-planar divisions increased to 54% (44/81) (*Figure 4K,L*). We also observed signs of multipolar spindles in *Diaph3* cKO tissue (*Figure 4M,N*), a phenotype that is reminiscent of DIAPH3 and SPAG5 knockdown (*Figure 1*, *Figure 3*). Finally, to determine whether the changes in spindle orientation correlate with cell fate, we assessed cell cycle exit and neurogenesis by BrdU injection at e13.5 and performed the analysis of BrdU+, Ki67+, Ki67−, and Tbr1+ cell numbers at e14.5 (*Figure 4O–S*). The fraction of proliferating cells (BrdU+Ki67+/BrdU+) was reduced (−15%, p=0.011; *Figure 4P*), whereas that of precursors which exited the cell cycle (BrdU+Ki67−/BrdU+) was enhanced in *Diaph3* cKO compared with controls (+20%, p=0.011; *Figure 4Q*). Accordingly, the ratio of non-cycling to cycling cells (BrdU+Ki67−/BrdU+Ki67+) was increased in *Diaph3* cKO (+40%, p=0.0078; *Figure 4R*), and so was the fraction of cells that underwent neurogenic division (BrdU+Tbr1+/BrdU+: +15%, p=0.0031; *Figure 4—figure supplement 1S*). Taken together, these results show that the lack of DIAPH3 not only causes the loss of progenitors, but also modifies their fate. Both cell death and fate decision defects disrupt neurogenesis.

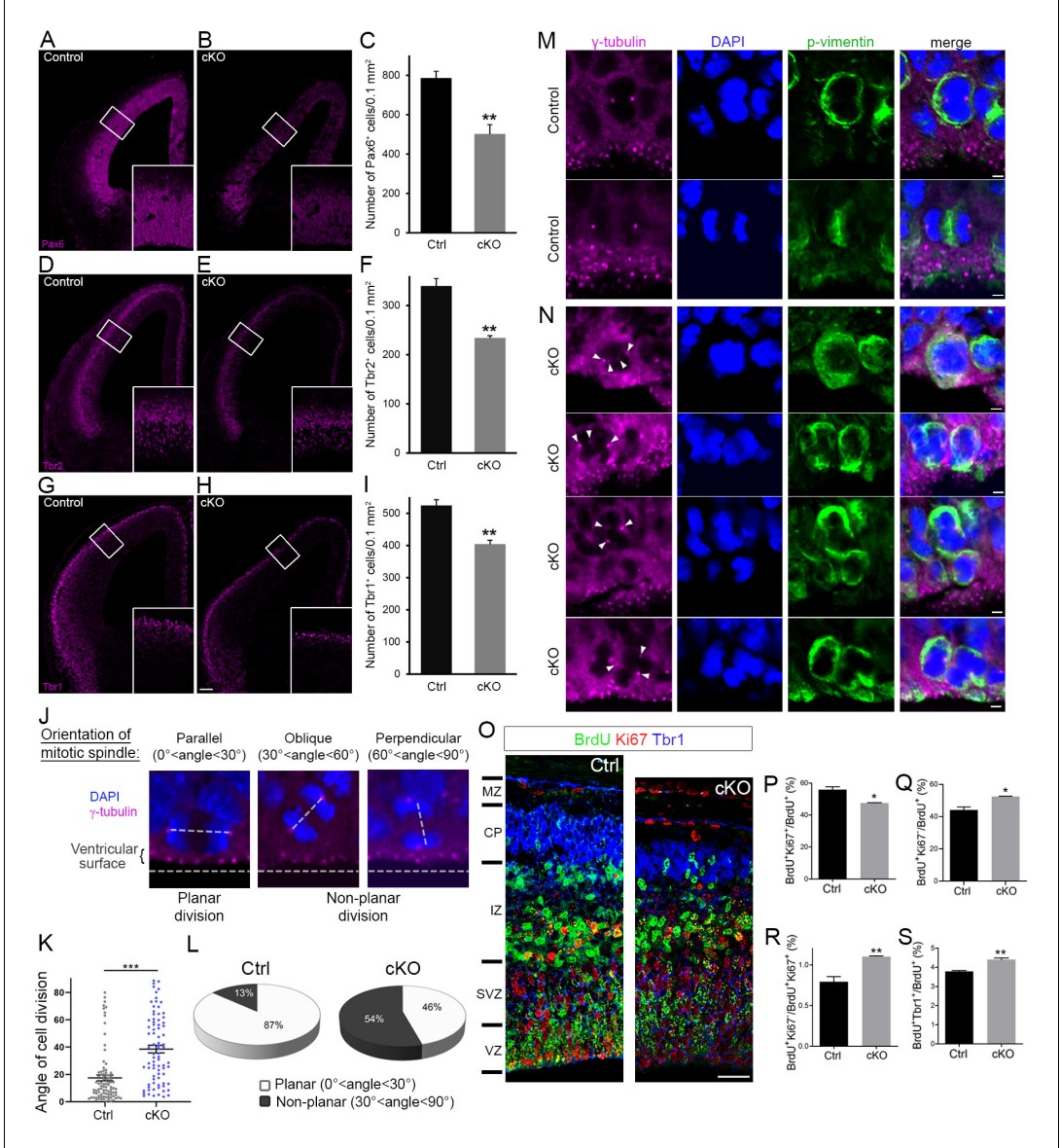

**Figure 4.** Cortex-specific inactivation of *Diaph3* disrupts cortical neurogenesis. (A, B, D, E, G, and H) Forebrain coronal sections from e13.5 stained with Pax6 (apical neural progenitor cells [aNPCs]; A, B), Tbr2 (basal progenitors; D, E), and Tbr1 (neurons; G, H) antibodies. Insets are enlargements of the boxed areas. Quantifications shown in (C, F, and I) emphasize reduction in the number of apical radial glia (aNPC, Pax6[+], p=0.00853), basal progenitors (Tbr2[+], p=0.00287), and neurons (Tbr1[+], p=0.00535), respectively. Cells were counted in 0.1 mm$^2$ cortical areas. $n$ = 3 embryos per genotype. Student's *t*-test; error bars represent s.e.m. Scale bar, 100 μm. (J) e12.5 cortical sections stained with anti-γ tubulin antibodies (magenta) and DAPI (blue) to label centrosomes and chromosomes, respectively, and 'foresee' the mitotic spindle. The orientation of the mitotic spindle with respect to the ventricular surface is categorized into three types: parallel (left), oblique (middle), or perpendicular (right). When the mitotic spindle is parallel to the ventricular surface (0°≤angle<30°), the cell division is planar. When it is oblique (30°≤angle<60°) or perpendicular (60°<angle<90°) to the ventricular surface, the division is non-planar. (K, L) Assessment of cell division modality at e12.5. Compared with control mice, there is a significant shift toward larger angles (p=5.9 × 10$^{-9}$, $n$ = 89 mitoses from three control embryos, 81 mitoses from three *Diaph3* cKO embryos; Student's *t*-test; error bars represent s.e.m.) (K) and an increase in the ratio of non-planar division in *Diaph3* cKO (L). (M) Dividing aNPCs in control telencephalic VZ stained with anti-γ tubulin antibodies (magenta), anti-phospho vimentin antibodies (green), and DAPI (blue) to label centrosomes, dividing cells, and chromosomes, respectively. (N) Illustrations of supernumerary centrosomes (arrowheads) in *Diaph3* cKO. Scale bar, 2 μm. (O) Assessment of proliferative versus neurogenic division (cell cycle exit) at e.14.5. Pregnant females were injected with BrdU at e.13.5 and embryos were collected after 24 hr. Forebrain coronal sections were processed for triple immunostaining using anti-BrdU (green), anti-Ki67 (red), and anti-Tbr1 (blue). BrdU[+], Ki67[+], and Tbr1[+] cells were counted in 200-μm-wide cortical stripes. (P–S) Quantification of proliferating cells (P, BrdU[+]Ki67[+]/BrdU[+]) and non-proliferating cells (Q, BrdU[+]Ki67[-]/BrdU[+]) shows that cell proliferation is reduced, whereas cell cycle exit is enhanced in *Diaph3* cKO compared to control. The ratio of noncycling to cycling cells (BrdU[+]Ki67[-]/BrdU[+]Ki67[+]) is also increased in *Diaph3* cKO (R). The fraction of progenitors exiting the cell cycle and generating Tbr1[+] neurons is

*Figure 4 continued on next page*

Figure 4 continued

increased in *Diaph3* cKO. n = 3 embryos for each genotype; Student's *t* test, p=0.011 for (**P**), p=0.011 for (**Q**), p=0.0078 for (**R**), p=0.0031 for (**S**); error bars represent s.e.m. Scale bar, 30 μm. CP, cortical plate; IZ, intermediate zone; MZ, marginal zone; SVZ, subventricular zone; VZ, ventricular zone. The online version of this article includes the following source data and figure supplement(s) for figure 4:

**Source data 1.** Quantification of cell division modalities.
**Figure supplement 1.** Cell apoptosis in *Diaph3* cKO mice.
**Figure supplement 1—source data 1.** Quantification of cell death in the Diaph3 cKO cortex.

## *Diaph3* cKO mice exhibit cortical hypoplasia and impaired behavior

The cerebral cortex of adult *Diaph3* cKO mice was markedly thin (*Figure 5A,B*). Immunostaining with markers of different cortical layers (Cux1: layer II-III, Foxp2: layer VI) revealed a diminished number of cells along with a reduced thickness of cortex (*Figure 5C–F*). We then assessed the consequences of cortical hypoplasia in terms of behavior. *Diaph3* cKO mice displayed lower spontaneous

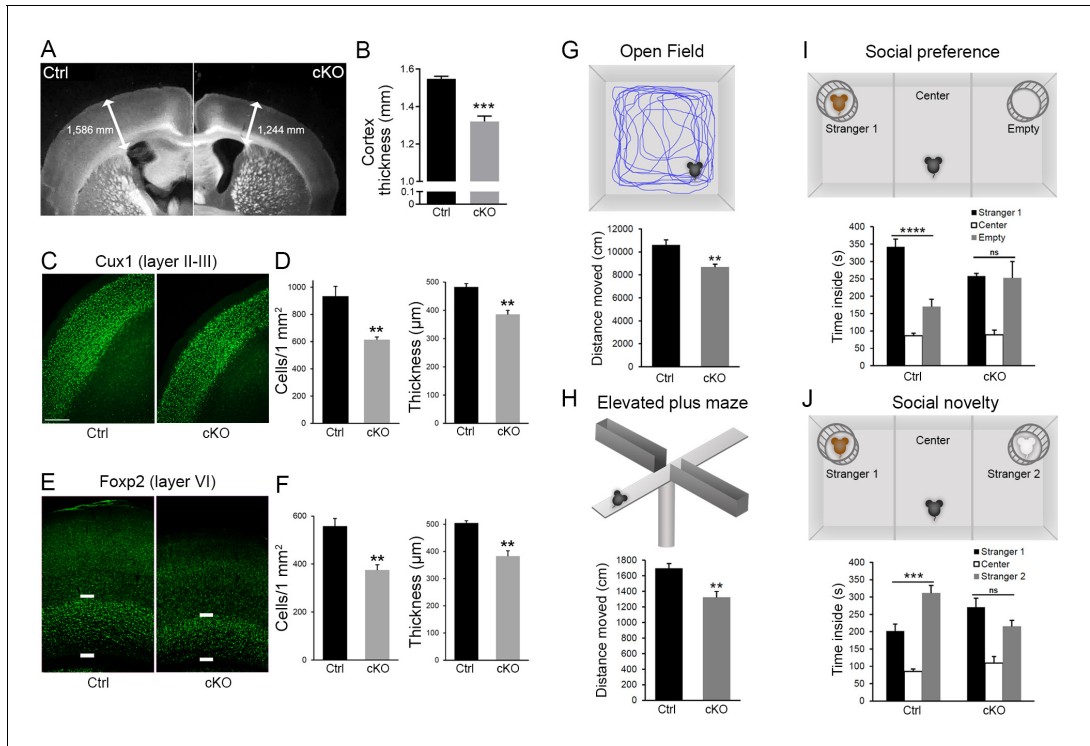

**Figure 5.** *Diaph3* cKO mice display cortical hypoplasia and behavioral defects. (**A**) Dark field micrograph of coronal sections of the forebrain from control (left) and *Diaph3* cKO (right) mice, depicting a marked reduction in cortex thickness in *Diaph3* cKO, quantified in (**B**) (1.3 ± 0.028 mm in *Diaph3* cKO versus 1.5 ± 0.014 mm in control, p=0.00038). (**C**) Coronal sections stained with the upper layer marker Cux1. (**D**) Quantification of the number of Cux1+ cells (left) and thickness (right) in layers II-III (Cux1+ cells: 614 ± 20 cells in *Diaph3* cKO versus 933 ± 73 in control, p=0.0056; thickness: 385 ± 15 μm in *Diaph3* cKO versus 482 ± 13 μm in control, p=0.0025). (**E**) Coronal sections stained with deep layer marker Foxp2. (**F**) Quantification of the number of Foxp2+ cells (left) and thickness (right) in layer VI (Foxp2+ cells: 375 ± 22 cells in *Diaph3* cKO versus 558 ± 32 in control, p=0.0033; thickness: 383 ± 20 μm in *Diaph3* cKO versus 505 ± 8 μm in control, p=0.0012). Cux1+ and Foxp2+ cells were counted in 1 mm² cortical area. n = 4 mice for each genotype, Student's *t*-test; error bars represent s.e.m. Scale bar, 200 μm. (**G**) Distance moved by *Diaph3* cKO and control mice in the open field. Student's *t* test, p=0.0015. (**H**) Distance travelled by mice in elevated plus maze. Student's *t* test, p=0.0043. (**I**) Social behavior in the 'three-chamber' test. One-way ANOVA test, p=8.4 × 10⁻⁷ for control, p=0.99 for *Diaph3* cKO (stranger versus empty chamber). (**J**) Social novelty behavior in the 'three-chamber' test. One-way ANOVA test, p=0.00050 for control, p=0.16 for *Diaph3* cKO (stranger 1 versus stranger 2 chamber). Compared with control, *Diaph3* cKO mice have reduced locomotor activity and defective social interactions. n = 10 per genotype. Error bars represent s.e.m.

The online version of this article includes the following source data and figure supplement(s) for figure 5:

**Source data 1.** Cortical histogenesis and behaviour.
**Figure supplement 1.** Behavior assessment of *Diaph3* cKO mice.
**Figure supplement 1—source data 1.** Assessement of learning and memory, and olfactory behaviour.

locomotor activity in open-field test and elevated plus maze (*Figure 5G,H*), but no difference was detected when we analyzed the time spent in the periphery of the open field or in the closed arms of the elevated plus maze (*Figure 5—figure supplement 1A,B*). These results suggest that DIAPH3 deficiency impairs locomotor activity, but has little effect, if any, on anxiety and/or attention. In addition, *Diaph3* cKO mice exhibited a defective social behavior in the 'three-chamber' test. They spent more time in the empty chamber than with a stranger mouse (stranger 1), emphasizing the lower sociability of *Diaph3* cKO mice, as compared to control littermates (*Figure 5I*). When another stranger mouse was introduced, *Diaph3* cKO mice spent less time with the new stranger, indicating impairment in social novelty behavior (*Figure 5J*). *Diaph3* cKO mice did not show any olfactory defect (*Figure 5—figure supplement 1C*). They performed as well as controls in the Morris water maze (*Figure 5—figure supplement 1D*) and modified Y-maze tests (*Figure 5—figure supplement 1E*), implying that long-term and short-term memories were preserved. Finally, they did not display any repetitive behavior, as assayed by self-grooming (*Figure 5—figure supplement 1F*) or marble burying (*Figure 5—figure supplement 1G*). These results show that *Diaph3* cKO mice have specific deficits in motor activity and social behavior, which is in line with previous studies that have associated DIAPH3 mutations with autism spectrum disorder in humans (*Vorstman et al., 2011*; *Xie et al., 2016*).

## Discussion

DIAPH3 is an effector of Rho GTPases that has been classically associated with actin dynamics in vitro (*Watanabe et al., 2010*; *Watanabe et al., 2013*). Functional analysis of knockout mice showed that it regulates cytokinesis in erythroid cells by promoting the accumulation of actin in the cleavage furrow at telophase (*Watanabe et al., 2013*). In addition to this well-documented role in actin cytoskeleton and cytokinesis, we have reported that DIAPH3 plays a role during karyokinesis and that its loss causes chromosome mis-segregation, aneuploidy, and cell death (*Damiani et al., 2016*). Here we provide evidence that DIAPH3 localizes to the centrosome during mitosis. Its depletion compromises the integrity of centrosome, spindle and astral MT, as well as the expression or stability of several proteins involved in spindle-cell cortex and spindle-kinetochore interactions (*Figure 6*). These results are consistent with previous in vitro data, suggesting that DIAPH3 may serve as a scaffold protein that binds to and stabilizes MT (*Palazzo et al., 2001*; *Bartolini et al., 2008*; *Wen et al., 2004*). By regulating centrosome integrity, microtubule stability, and spindle orientation, DIAPH3 sits at the heart of cell division accuracy and fate determination (*Figure 6*). All these processes are requisites for the production and maintenance of neural progenitors and neurons. The centrosome acts as a seed and anchoring point for the minus end of MT as they grow toward the equatorial region (spindle MT) or the cell cortex (astral MT). Consistent with this, we found that depletion of DIAPH3 disrupts the centrosome and results in disorganized spindle and astral MT, abnormal alignment, and segregation of chromosomes. DIAPH3 depletion affects the expression level and/or distribution of several proteins. Among those, SPAG5, which localizes at the centrosome and kinetochore, is essential to the recruitment of other centrosomal proteins such as CDK5RAP2 (MCPH3) (*Kodani et al., 2015*). SPAG5 controls sister chromatid cohesion by regulating separase activity (*Thein et al., 2007*). Knockdown of SPAG5 induces similar mitotic errors to those triggered by knockdown of DIAPH3 (*Kodani et al., 2015*; *Thein et al., 2007* and this work), and its overexpression rescues DIAPH3 knockdown phenotype. DIAPH3 depletion also affects the expression level or the stability of GPSM2, NUMB, and PAR3, all of which localize at the region of the cell cortex where astral MT are anchored.

Microcephaly is a reduction of more than three standard deviations of the head circumference compared to matching gender, age, and ethnicity controls (*Kaindl et al., 2010*; *Jayaraman et al., 2018*). Primary microcephaly is an inherited neurodevelopmental defect that appears mostly during pregnancy or at the first postnatal year. Eighteen 'microcephaly' genes named MCPH 1–18 (for microcephaly primary hereditary 1–18) have been identified (*Jayaraman et al., 2018*). All these genes are expressed in the germinal zones, wherein the neural progenitors reside, proliferate, and differentiate into neurons during the second trimester of pregnancy (*Verloes et al., 1993*; *Gilmore and Walsh, 2013*). A key determinant of brain size is the production and maintenance of the right numbers of neural progenitors and neurons. To attain the initial pool of progenitors and right number of neurons, NSCs must reconcile two imperatives: the high speed and fidelity of cell

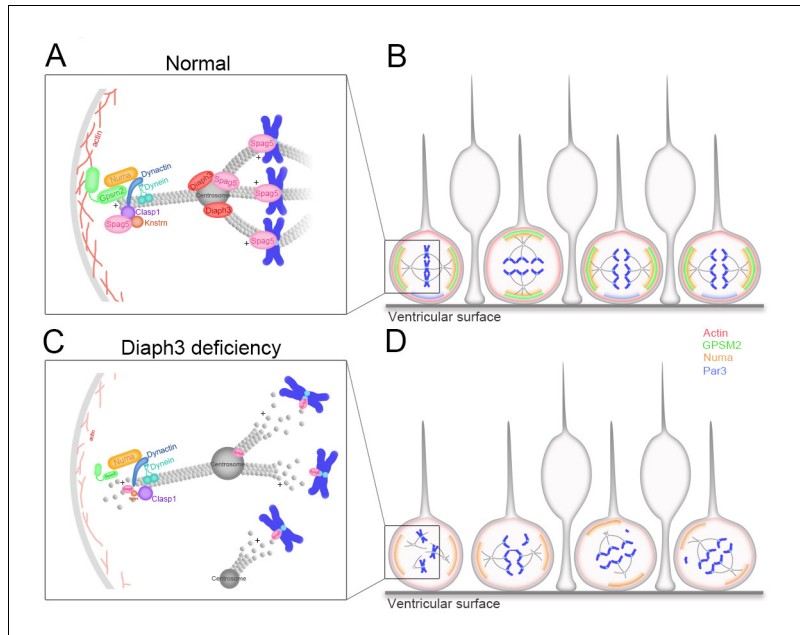

**Figure 6.** Working model of DIAPH3 function in aNPC. (**A**) During mitosis, diaphanous three (DIAPH3) maintains the dynamics of cytoskeleton. Polarity proteins nuclear mitotic apparatus (NUMA) and G-protein signaling modulator 2 (GPSM2) assemble underneath cortical F-actin and recruit the dynein/dynactin motor protein complex. Dynein and dynactin interact with SPAG5/KNSTRN/CLASP1 located at the microtubule plus-end and attach astral microtubule (MT) to cell cortex, thus providing the pulling force for chromosome bipolar segregation (*Okumura et al., 2018*; *Dunsch et al., 2011*; *Kern et al., 2016*). (**B**) The position of polarity proteins NUMA/ GPSM2 directs the orientation of the mitotic spindle and determines the type of division (proliferative versus neurogenic) of apical neural progenitor cells (aNPCs). (**C, D**) Absence of DIAPH3 destabilizes actin and MT and disrupts the expression of SPAG5, KNSTRN, GPSM2, PAR3, and NUMB, therefore weakening astral MT-cell cortex and spindle MT-kinetochore interactions. This causes spindle abnormalities, chromosome mis-alignment, and mis-segregation, and alters the fate decision of aNPCs.

division. This relies on the interplay between cell cycle regulators, mitotic spindle assembly, mitotic checkpoints, and cytoskeleton. Hence, the division of NSCs and the balance between proliferation and differentiation of apical progenitors is a critical factor of brain size (*Gilmore and Walsh, 2013*). Defects in the balance between proliferative and neurogenic division, or in the timing of neurogenic switch, affect cell fate and production of neurons. Loss of function of DIAPH3 disrupts the modality of cell division and fate determination of apical progenitors, enhancing the number of neurogenic divisions at the expense of proliferative division, prematurely exhausting the pool of progenitors. Even though the cytocortical proteins help to orientate the mitotic spindle, the loss of GPSM2 does not lead to microcephaly despite the randomized cleavage planes in dividing neural progenitors (*Konno et al., 2008*; *Blumer et al., 2008*). Therefore, the role of DIAPH3 in fidelity of nuclear division and the cell loss observed in *Diaph3* cKO mice may be more instrumental to the emergence of microcephaly in *Diaph3* cKO mice than its role in oriented cell division. It will be interesting to test whether the blockade of apoptosis induces neoplastic transformation due to accumulation of aneuploid cells. Finally, DIAPH3 is expressed in several epithelia and may govern cell proliferation, thereby influencing the size of different organs and/or organisms. In support of this, patients with microcephalic osteodysplastic primordial dwarfism type 1 (MOPD 1) syndrome display a 50% reduction in DIAPH3 expression (*Edery et al., 2011*). Furthermore, the *Diaph3* gene has been associated with body size in equines, as copy number variants were reported in donkeys, ponies, and horses with short stature (*Metzger et al., 2018*).

In humans, two clinical studies point to *DIAPH3* as an autism susceptibility gene (*Vorstman et al., 2011*; *Xie et al., 2016*). The first study identified a double-hit mutation with one amino acid substitution (Pro614Thr) in one allele and a deletion in the second allele. The second is a de novo point mutation in the 18th exon (c.2156T > C, pI719T). Several chromosomal deletions spanning the

*DIAPH3* locus have also been associated with language impairment, autism, and intellectual disability (https://decipher.sanger.ac.uk/ and *Nathalie Sans et al., 2016*). Importantly, among autistic patients, a significant fraction exhibits microcephaly (*Fombonne et al., 1999*). In this work, we report that the conditional deletion of *Diaph3* in the mouse cerebral cortex dramatically affects neurogenesis and leads to microcephaly and autistic-like behavior, with typical altered motor activity and social interactions. Our finding that the deletion of *Diaph3* affects the accuracy and modalities of division of neural progenitor cells, ultimately regulating the final number of cortical neurons, may provide a molecular and cellular basis to the brain malformations and neurological disorders that have been associated with DIAPH3 dysfunction in humans.

Combined together, our results suggest a novel role of DIAPH3 as a centrosomal protein. Its deficiency induces centrosome abnormalities and disrupts spindle and astral MT. These defects cause inaccuracies in nuclear division and fate decision of neural progenitors, leading to the loss of cortical progenitors, microcephaly, and autistic-like behavior.

# Materials and methods

**Key resources table**

| Reagent type (species) or resource | Designation | Source or reference | Identifiers | Additional information |
|---|---|---|---|---|
| Antibody | Anti-CDP/Cux1 (rabbit polyclonal) | Santa Cruz | Cat# SC-13024, RRID:AB_2261231 | IF (1:200) |
| Antibody | Anti-Foxp2 (rabbit polyclonal) | Abcam | Cat# ab16046, RRID:AB_2107107 | IF (1:500) |
| Antibody | Anti-Tbr1 (rabbit polyclonal) | Abcam | Cat# ab31940, RRID:AB_2200219 | IF (1:500) |
| Antibody | Anti-Tbr2 (rabbit polyclonal) | Abcam | Cat# ab23345, RRID:AB_778267 | IF (1:500) |
| Antibody | Anti-Pax6 (rabbit polyclonal) | Covance | Cat# PRB-278P, RRID:AB_291612 | IF (1:500) |
| Antibody | Anti-γ-tubulin (rabbit polyclonal) | Abcam | Cat# ab11317, RRID:AB_297921 | IF (1:500) |
| Antibody | Anti-Diaph3 (rabbit polyclonal) | *Tominaga et al., 2000* | | IF (1:500) WB (1:500) |
| Antibody | Anti-Spag5 (rabbit polyclonal) | Proteintech | Cat# 14726–1-AP, RRID:AB_2194787 | IF (1:200) |
| Antibody | Anti-α-tubulin (mouse monoclonal) | Sigma | Cat# T6199, RRID:AB_477583 | IF (1:500) |
| Antibody | Anti-cleaved caspase-3 (Alexa Fluor 488 conjugate) (rabbit monoclonal) | Cell Signaling | Cat# 9603, RRID:AB_11179205 | IF (1:100) |
| Antibody | Anti-Ki67 (mouse monoclonal) | BD phamingen | Cat# 556003, RRID:AB_396287 | IF (1:500) |
| Antibody | Anti-SPAG5 (rabbit polyclonal) | Sigma | Cat# HPA022008, RRID:AB_1853519 | WB (1:750) |
| Antibody | Anti-GAPDH (chicken polyclonal) | Millipore | Cat# AB2302, RRID:AB_10615768 | WB (1:2000) |
| Antibody | Anti-NUMA (mouse monoclonal) | BD Biosciences | Cat# 610561, RRID:AB_397913 | WB (1:500) |
| Antibody | Anti-PAR3 (rabbit polyclonal) | Millipore | Cat# 07–330, RRID:AB_2101325 | IF (1:500) WB (1:500) |
| Antibody | Anti-GPSM2 (rabbit polyclonal) | *Ezan et al., 2013* | | IF (1 :200) WB (1 :200) |

*Continued on next page*

*Continued*

| Reagent type (species) or resource | Designation | Source or reference | Identifiers | Additional information |
|---|---|---|---|---|
| Antibody | Anti-CENPA (rabbit monoclonal) | Cell Signaling | Cat# 2048, RRID:AB_1147629 | WB (1 :500) |
| Antibody | Anti-Dynein IC1/2 (mouse monoclonal) | Santa Cruz | Cat# sc-13524, RRID:AB_668849 | WB (1:500) |
| Antibody | Anti-KNSTRN (rabbit polyclonal) | Sigma | Cat# HPA042027, RRID:AB_10797378 | IF (1:1000) WB (1:1000) |
| Antibody | Anti-INSC (rabbit polyclonal) | Abcam | Cat# ab102953, RRID:AB_10711784 | WB (1:1000) |
| Antibody | Anti-NUMB (goat polyclonal) | Abcam | Cat# ab4147, RRID:AB_304320 | IF (1:100) WB (1:100) |
| Antibody | Anti-p150Glued /dynactin (mouse polyclonal) | BD Biosciences | Cat# 610474, RRID:AB_397846 | WB (1:500) |
| Antibody | Anti-CLASP1 (rabbit monoclonal) | Abcam | Cat# ab108620, RRID:AB_10864427 | WB (1:5000) |
| Antibody | Anti-BrdU (rat monoclonal) | Serotec | Cat# MCA2060GA, RRID:AB_10545551 | IF (1:200) |
| Antibody | Anti-Ki67 (rabbit polyclonal) | Abcam | Cat# ab15580, RRID:AB_443209 | IF (1:250) |
| Antibody | Anti-Tbr1 (chicken polyclonal) | Millipore | Cat# AB2261, RRID:AB_10615497 | IF (1:100) |
| Antibody | Anti-SPAG5 (rabbit polyclonal) | NovusBio | Cat# NB100-74638, RRID:AB_2239831 | WB (1:1000) |
| Antibody | Alexa Fluor 488 anti-mouse IgG (goat polyclonal) | Invitrogen | Cat# A11017, RRID:AB_143160 | IF (1:800) |
| Antibody | Alexa Fluor 488 anti-rabbit IgG (goat polyclonal) | Invitrogen | Cat# A11034, RRID:AB_2576217 | IF (1:1000) |
| Antibody | Alexa Fluor 488 anti-chicken IgY (IgG) (goat polyclonal) | Jackson ImmunoResearch | Cat# 103-545-155, RRID:AB_2337390 | IF (1:1000) |
| Antibody | Alexa Fluor 568 anti-mouse IgG (goat polyclonal) | Invitrogen | Cat# A21124, RRID:AB_141611 | IF (1:1000) |
| Antibody | Alexa Fluor 568 anti-rabbit IgG (goat polyclonal) | Invitrogen | Cat# A11036, RRID:AB_10563566 | IF (1:1000) |
| Antibody | Anti-chicken IgY/HRP-linked (rabbit polyclonal) | Upstate Biotech | Cat# 12–341, RRID:AB_390189 | WB (1:1000) |
| Antibody | Anti-mouse IgG/HRP-linked (goat polyclonal) | Dako | Cat# P0447, RRID:AB_2617137 | WB (1:1000) |

*Continued on next page*

*Continued*

| Reagent type (species) or resource | Designation | Source or reference | Identifiers | Additional information |
|---|---|---|---|---|
| Antibody | Anti-rabbit IgG/HRP-linked (goat polyclonal) | Cell Signaling | Cat# 7074, RRID:AB_2099233 | WB (1:1000) |
| Cell line (*Homo-sapiens*) | U2OS cell (osteosarcoma) | ATCC | Cat# HTB-96, RRID:CVCL_0042 | |
| Cell line (*Mus musculus*) | NIH3T3 immortalized MEFs | ATCC | Cat# CRL-1658 RRID:CVCL_KS54 | |
| Strain, strain background | *Mus musculus* | Jackson Lab | *Emx1^{tm1(cre)Krj}*/J RRID:IMSR_JAX:005628 | |
| Strain, strain background | *Mus musculus* | This paper | *Diaph3^{Emx1-Cre}* cKO (*Diaph3^{f/f}*; *Emx1-Cre*) | |
| Transfected construct (human) | Scrambled shRNA | Origene | Cat# TR30012 | Transfected construct |
| Transfected construct (human) | sh-Diaph3 | Origene | Cat# TR304992 | Transfected construct |
| Transfected construct (human) | sh-Spag5 | Origene | Cat# TR309161 | Transfected construct |
| Transfected construct (*Mus musculus*) | sh-Spag5 | Origene | Cat# TR509034 | Transfected construct |
| Commercial assay or kit | Mycoplasma PCR detection kit | Sigma | Cat#, MP0035 | |
| Commercial assay or kit | RNeasy Kit | Qiagen | Cat# 74004 | |
| Commercial assay or kit | RT cDNA synthesis Kit | Promega | Cat# A5003 | |
| Commercial assay or kit | SYBR Green SuperMix | Biorad | Cat# 170–8882 | |
| Commercial assay or kit | Lipofectamine LTX | Thermofisher Scientific | Cat# 15338030 | |
| Commercial assay or kit | RNAscope Probe-Mm-Spag5 | ACD | Cat# 505691 | |
| Commercial assay or kit | Pierce BCA protein assay kit | Thermofisher Scientific | Cat# 23225 | |
| Commercial assay or kit | Bolt 4–12% Bis-Tris Plus Gels | Thermofisher Scientific | Cat# NW04125 BOX | |
| Commercial assay or kit | StartingBlock T20 (TBS) blocking buffer | Thermofisher Scientific | Cat# 37543 | |
| Commercial assay or kit | 20x Bolt MOPS SDS running buffer | Thermofisher Scientific | Cat# B0001 | |
| Software, algorithm | Prism | GraphPad, USA | RRID:SCR_002798 | |
| Software, algorithm | Zen lite | Zeiss | | |
| Software, algorithm | Ethovision 6.1, Noldus | Wageningen, The Netherlands | | |

## Mutant mice

All animal procedures were carried out in accordance with European guidelines and approved by the animal ethics committee of the Université Catholique de Louvain. Mouse lines used in this study were *Emx1^{tm1(cre)Krj}*/J; (Jackson Lab) (*Gorski et al., 2002*; *Damiani et al., 2016*), *Diaph3^{f/f}*; *Emx1-Cre*

(*Diaph3* cKO), and *Diaph3*^{f/f} (control for *Diaph3* cKO). All mice were maintained in a mix background.

## Immunostaining and antibodies

For immunohistochemistry, embryos were fixed in 4% paraformaldehyde (PFA), cryoprotected by gradients of sucrose solution. Cryosections were blocked in PBST (0.1% Triton X-100 in phosphate-buffered saline [PBS]) supplemented with 5% normal goat serum and 1% bovine serum albumin (BSA) for 30 min. Slides were incubated in primary antibodies diluted in blocking buffer at 4°C overnight. Slides were washed and incubated with secondary antibodies (Alexa, Thermofisher) diluted in the same blocking buffer. Cultured cells were fixed with chilled methanol for 5 min, followed by washing in PBST. Cells were incubated in blocking buffer for 30 min, primary antibodies for 1.5 hr, and secondary antibodies for 1.5 hr at room temperature. Primary antibodies used were as follows: rabbit anti-Cux1 (Santa Cruz, SC-13024, 1:200), rabbit anti-Foxp2 (Abcam, ab16046, 1:500), rabbit anti-Tbr1 (Abcam, ab31940, 1:500), rabbit anti-Tbr2 (Abcam, ab23345, 1:500), rabbit anti-Pax6 (Covance, PRB-278P, 1:500), rabbit anti-γ-tubulin (Abcam, ab11317, 1:500), Diaph3 (1:500) (*Tominaga et al., 2000*), rabbit anti-Spag5 (Proteintech, 14726–1-AP, 1:200), mouse anti-α-tubulin (Sigma, T6199, 1:500), rabbit anti-aCas3 (Cell Signaling, 9603, 1:100), and mouse anti-Ki67 (BD Pharmingen, 556003, 1:500). Nuclei/chromosomes were counterstained with DAPI. Images were acquired with an Olympus FV1000 confocal microscope. Mitotic cells were imaged with Z-stack, which focused on the levels of centrosomes.

## Cell fate assessment

For cell fate and neurogenesis study, pregnant females were intraperitoneally injected with BrdU (Sigma B5002) (50 mg/kg body weight) at E13.5. Embryos were collected 24 hr later (E14.5) and brains were fixed in PFA 4% and processed for cryosectioning (20 μm). For BrdU staining, sections were pretreated with HCl 1 N for 20 min at room temperature, followed by a 10 min incubation with 0.1 M sodium borate buffer (pH 8.5). After that, antigen retrieval was performed by heating sections for 20 min in 0.01 M sodium citrate buffer (pH 6). Immunodetection was done using primary antibodies rat anti-BrdU (1:200, Serotec MCA2060GA), rabbit anti-Ki67 (1:250, Abcam ab15580), and chicken anti-Tbr1 (1:100, Millipore AB2261) and secondary antibodies as described above.

## Western blotting

Tissues were homogenized in lysis buffer containing 50 mM Tris HCl (pH 7.5), 150 mM NaCl, 1% NP40, and protease inhibitors (Roche). Cell lysates were incubated on ice for 30 min prior to centrifugation at 2000 g for 10 min at 4°C. Protein quantification was performed with BCA Protein Assay kit (Pierce). Supernatants were mixed with SDS-loading buffer and heated at 85°C for 10 min. Equal amounts of proteins were loaded on 4–12% gel (Invitrogen) and transferred to PVDF membrane (Merck Millipore). Membranes were blocked with StartingBlock buffer (ThermoScientific) and incubated overnight at 4°C with rabbit anti-DIAPH3 (1:5000) (*Tominaga et al., 2000*), rabbit anti-SPAG5 (Sigma, HPA022008, 1:750), chicken anti-GAPDH (Millipore, AB2302, 1:2000), mouse anti-NUMA (Becton Dickinson, 610561, 1:500), rabbit anti-PAR3 (Millipore, 07–330, 1:500), rabbit anti-GPSM2 (1:200) (*Ezan et al., 2013*), rabbit anti-CENPA (Cell Signaling, 2048, 1:500), mouse anti-dynein (Santa Cruz, sc-13524, 1:500), rabbit anti-KNSTRN (Sigma, HPA042027, 1:1000), rabbit anti-INSC (Abcam, ab102953, 1:1000), goat anti-NUMB (Abcam, ab4147, 1:100), mouse anti-dynactin (BD Biosciences, 610474, 1:500), and rabbit anti-CLASP1 (Abcam, ab108620, 1:5000). Proteins were detected with SuperSignal West Pico PLUS solution (ThermoScientific) or SuperSignal West Femto Maximum Sensitivity Substrate kit (ThermoScientific). Band intensity was imaged by fusion pulse and quantified with ImageJ Gel analyzer tool. Values were normalized with GAPDH. The amount of protein in control was set to one.

## qPCR

Total mRNA was isolated from control or *Diaph3* cKO E11.5 telecephalon using the RNeasy mini kit (Qiagen) according to the supplier's instructions. Reverse transcription was performed with an RT cDNA Synthesis Kit (Promega). Real-time PCR was performed with SYBR green SuperMix using an iCycler real-time PCR detection system (Bio-Rad).

## Cell lines and knockdown experiments

U2OS and NIH/3T3 cell lines were purchased from ATCC. Their identity has been authenticated by STR profiling (ATCC). They were tested negative for mycoplasma using the 'LookOut Mycoplasma PCR Detection Kit' (Sigma, MP0035). Cells were cultured in Dulbecco's modified eagle's medium (DMEM) supplemented with penicillin-streptomycin and 10% fetal bovine serum (FBS; Invitrogen). Cells were seeded in 12-well plates (day 0) and were transfected with Lipofectamine LTX (Invitrogen) (day 1) using scrambled shRNA (Origene, TR30012) or pool of sh-DIAPH3 (Origene, TR304992) with sequences ATTTATGCGTTGTGGATTGAAAGAGATAT, AATCAGCATGAGAAGA TTGAATTGGTTAA, CACGGCTCAGTGCTATTCTCTTTAAGCTT, and AAGAGCAGGTGAACAACA TCAAACCTGAC or pool of sh-Spag5 (Origene, TR309161) with sequences CCTCAAGGACACTG TAGAGAACCTAACGG, GGTAGGATTCTTGGCTCTGATACAGAGTC, CTCCAAGGAAAGCCTGAG-CAGTAGAACTG, and AGATGAAGAGCCAGAATCAACTCCTGTGC. Cells were harvested 72 hr after transfection for western blotting. For immunostaining, cells were serum-deprived at day 3 (48 h after transfection) and seeded onto glass coverslips pre-coated with gelatin at day 4 (72 h after transfection). Cell cycle was synchronized by serum add back, and cells were fixed after 16 hr. For cell survival assay, cells were seeded at the same density (50% confluency, $2 \times 10^5$ cells per well) onto glass coverslips pre-coated with gelatin in 12-well plates at day 0. Cell number was counted at day 1 (just before transfection) and day 4 (72 h after transfection). Cells were then immunostained with apoptotic marker (aCas3), proliferation marker (Ki67), and DAPI. Cell density was imaged and counted by Zen lite with Zeiss AXIO light microscope.

## Behavioral tests

All the behavioral tests were performed using adult *Diaph3* cKO (*Diaph3^{f/f}; Emx1-Cre*) and control littermate (*Diaph3^{f/f}*) males. The 'open-field' test was performed to assess locomotor activity and anxiety. Mice were allowed to move freely in a square arena (60 × 60 cm) and video tracked (Ethovision 6.1, Noldus; Wageningen, The Netherlands) for 20 min (*Mallon et al., 2008*; *Mignion et al., 2013*). The total distance covered by test mice (locomotor activity) and the time spent in the periphery (anxiety) were measured.

The 'elevated plus maze' test was performed to assess locomotor activity and anxiety. Mice were placed in a setup consisting of two opposing open arms (exposed place) and two opposing closed arms (safer place). Total distance travelled and time spent in the closed arm were recorded by a video tracking system (Ethovision 6.1, Noldus; Wageningen, The Netherlands) for 5 min.

The 'three-chamber' test was performed to assess sociability. Test box was divided into three equal compartments of 20 cm each, and dividing walls had retractable doorways allowing access into each chamber. Mice were habituated in the middle chamber for 5 min with the doorways closed. To evaluate social interactions, mice were enclosed in the center compartment, and an unfamiliar mouse (called stranger 1) was restricted in a wire cage placed in one side compartment, while the other side compartment contained an empty wire cage. Mice were video tracked (Ethovision 6.1, Noldus; Wageningen, The Netherlands) for 10 min. The time spent in each chamber was measured. To evaluate the preference for social novelty, a second unfamiliar mouse (called stranger 2) was introduced into the empty wire cage in the other side compartment of the sociability test box. Mice were video tracked for 10 min, and the time spent in each chamber was measured (*Moy et al., 2007*).

The 'food localization' test was performed to evaluate the olfactory function. Test mice were fasted for 16 hr with water supply ad libitum. They were then transferred into a clean cage with 3-cm-thick wood-chip bedding and lightly tamped down to make a flat surface. 1.0 g of food pellet was placed at the same location under the bedding and mice were introduced into the cage at a constant position. The time from which the mouse was placed into the cage until it retrieved the food pellet with its front paws was counted up to a maximum of 300 s.

The 'Morris water maze' test was performed to assess long-term memory. Water maze was made of a round pool with a diameter of 113 cm, virtually divided into four quadrants (North, South, West, and East) and filled with water (26°C). Several visual cues were placed around the pool. The platform was placed at the center of the North-East quadrant of the pool and maintained in this position throughout the four days. Mice were video-tracked (Ethovision 6.1, Noldus; Wageningen, The

Netherlands). The time latency to reach the platform was measured (*Rzem et al., 2015*; *Boucherie et al., 2018*).

The 'Y-maze' test was performed to assess short-term memory (*Rzem et al., 2015*; *Lepannetier et al., 2018*). The Y-maze was made of three identical opaque arms. The test mouse was freely accessible to only two arms for 10 min. After a 30 min inter-trail interval, the third arm was opened and the mouse was put back into the maze. The mouse was video tracked, and the time spent in the novel arm was assessed.

'Self-grooming' and 'marble burying' tests were performed to assess repetitive behavior. For self-grooming assessment, mice were individually placed in a clear plastic cage for 20 min. The first 10 min served as a habituation period, and during the second 10 min of testing, time spent grooming was recorded. For the marble burying test, test cages were filled with wood-chip bedding and lightly tamped down to make a flat surface. Twenty glass marbles were placed on the surface in a regular pattern and mice were then placed in the cage. After 30 min, the number of bedding-buried marbles was counted (*Amodeo et al., 2012*).

## In utero electroporation

CD1 pregnant females were used for in utero electroporation (IUE). e13.5 embryos were electroporated with 1 µg/µl of scrambled shRNA or sh-Spag5, and 0.5 µg/µl of CAG-GFP in 10 mM Tris buffer (pH 8.0) and 0.01% fast green as described in *Jossin and Cooper, 2011*. Needles for injection were pulled from Wiretrol II glass capillaries (Drummond Scientific). Forcep-type electrodes (Nepagene) with 5 mm pads were used for electroporation using the ECM830 electroporation system (Harvard Apparatus). Embryos were collected and fixed at e15.5. shRNA were purchased from Origene; scrambled shRNA (Origene, TR30012) and pool of sh-Spag5 (Origene, TR509034) with sequences TAGTCTCTGGAGACCTGTTGTCCTTGCTT, GGAGGAAGCAATAGAAACAGTGGATGACT, GACAAGTATCTGAGCCATAGGCACATCCT, and GCAACAAGGAGCAGGCTACTCAATGGCAA were used.

## In-situ hybridization

RNase-free coronal cryosections from e11.5 *Diaph3* cKO and control embryos were hybridized with Spag5 fast red-labelled RNAscope probe (Cat No. 505691) as described by the manufacturer.

## Acknowledgements

We thank Dr. Ulrike Gruneberg (University of Oxford, UK) for the Spag5 plasmid, Dr. Mireille Montcouquiol (INSERM, France) for the GPSM2/PINS antibody, and Valérie Bonte, Rachid El Kaddouri, Isabelle Lambermont, and Younes Massaoudi for technical support. This study makes use of data generated by the DECIPHER community. A full list of centers that contributed to the generation of the data is available from http://decipher.sanger.ac.uk. Funding for the DECIPHER project was provided by the Wellcome Trust. This work was supported by the following grants: FNRS PDR T00075.15, FNRS PDR T0236.20, FNRS-FWO EOS 30913351, Fondation Médicale Reine Elisabeth, and Fondation JED-Belgique. EOL was supported by 'Move-In Louvain' postdoctoral fellowship funded by Marie Skłodowska-Curie Actions of European Commission and Université Catholique de Louvain. RS is supported by an FNRS-Télévie Grant. GC, NRR, and YJ are research fellow, postdoctoral researcher, and research associate at the Belgian Fund for Scientific Research (FNRS), respectively.

## Additional information

### Competing interests

Fadel Tissir: Reviewing editor, *eLife*. The other authors declare that no competing interests exist.

### Funding

| Funder | Grant reference number | Author |
| --- | --- | --- |
| Fonds De La Recherche Scien- | | Fadel Tissir |

tifique - FNRS

| Fonds De La Recherche Scientifique - FNRS | FNRS PDR T0236.20 | Fadel Tissir |
| Fonds De La Recherche Scientifique - FNRS | EOS 30913351 | Fadel Tissir |
| H2020 Marie Skłodowska-Curie Actions | | Eva On-Chai Lau |
| Université Catholique de Louvain | | Eva On-Chai Lau |
| Fonds De La Recherche Scientifique - FNRS | | Rana Saade |
| Fonds De La Recherche Scientifique - FNRS | | Georges Chehade Nuria Ruiz-Reig Yves Jossin |

The funders had no role in study design, data collection and interpretation, or the decision to submit the work for publication.

## Author contributions
Eva On-Chai Lau, Investigation, Visualization, Methodology, Writing - original draft, Writing - review and editing; Devid Damiani, Conceptualization, Investigation, Methodology, Writing - review and editing, Contributed equally with Georges Chehade; Georges Chehade, Supervision, Validation, Investigation, Visualization, Writing - review and editing, Contributed equally with Devid Damiani; Nuria Ruiz-Reig, Validation, Investigation, Methodology, Writing - review and editing; Rana Saade, Mohamed Aittaleb, Formal analysis, Investigation; Yves Jossin, Validation, Investigation, Visualization; Olivier Schakman, Validation, Investigation, Writing - original draft; Nicolas Tajeddine, Supervision, Validation, Methodology, Writing - review and editing; Philippe Gailly, Conceptualization, Validation, Investigation, Writing - review and editing; Fadel Tissir, Conceptualization, Resources, Formal analysis, Supervision, Funding acquisition, Validation, Visualization, Methodology, Writing - original draft, Writing - review and editing

## Author ORCIDs
Nuria Ruiz-Reig (iD) https://orcid.org/0000-0001-7008-7920
Yves Jossin (iD) http://orcid.org/0000-0001-8466-7432
Fadel Tissir (iD) https://orcid.org/0000-0002-9292-6622

## Ethics
Animal experimentation: All animal procedures were carried out in accordance with European guidelines and approved by the animal ethics committee of the Université Catholique de Louvain (permit number 2019/UCL/MD/006).

## Decision letter and Author response
Decision letter https://doi.org/10.7554/eLife.61974.sa1
Author response https://doi.org/10.7554/eLife.61974.sa2

# Additional files

## Supplementary files
• Supplementary file 1. Cell division abnormalities in DIAPH3 knockdown cells. Over 81% of DIAPH3 knockdown cells exhibited mitotic errors (4.2% in control cells). These errors were divided into five 'non-exclusive' categories: (1) defects in chromosome alignment (57.4%); (2) abnormal astral microtubule (MT) (52.2%); (3) abnormal number of centrosomes (40%); (4) abnormal mitotic spindle (53%); and (5) mis-localization of SPAG5 (60%). $n$ = 118 cells transfected with scrambled shRNA and 115 with sh-DIAPH3 from five individual experiments.

• Supplementary file 2. Relative change in expression of proteins related to spindle orientation in *Diaph3 cKO.* n = 4 embryos for each genotype. Student's *t*-test, *p<0.05, **p<0.01.

• Transparent reporting form

### Data availability

All data generated or analysed during this study are included in the manuscript and supporting files. Source data files have been provided for all figures.

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
