## [Decision Letter]

**Acceptance summary:**

The paper uncovers critical functions of DIAPH3, providing data for its role in mitosis, cell survival and cell fate determination through interacts with the kinetochore-associated protein SPAG5.

**Decision letter after peer review:**

Thank you for submitting your article "DIAPH3 deficiency links microtubules to mitotic errors, defective neurogenesis, and brain dysfunction" for consideration by *eLife*. Your article has been reviewed by 3 peer reviewers, and the evaluation has been overseen by a Reviewing Editor and Jonathan Cooper as the Senior Editor. The reviewers have opted to remain anonymous.

The reviewers have discussed the reviews with one another and the Reviewing Editor has drafted this decision to help you prepare a revised submission.

This manuscript describes the role of Diaph3 in cortical development. Previous work from this group found that Diaph3 deletion in mice results in the loss of neural progenitors, cell death, and aneuploidy attributed to compromised spindle assembly checkpoint activation. Here they report that DIAPH3 does not only bind to actin network, as expected, but is also localized at the level of the centrosome. The major findings of this paper are: 1) depletion of DIAPH3 causes defects in centrosome formation/mitotic spindle orientation, resulting the mis-segregations of chromosomes; 2) depletion of DIAPH3 leads to the reduction in several other proteins associated with cell polarity and microtubule organization; 3) loss of SPAG5 phenocopies the loss of DIAPH3, and overexpression of SPAG5 can rescue some of the defects in DIAPH3 depleted cells; 4) conditional deletion of Diaph3 in the brain results in microcephaly and behavioral defects. There is concern that the current data are somewhat limited with regards to novel findings compared with the previous report (Damiani 2016), and the new findings on the links with centrosomal function and SPAG5 remain somewhat superficial.

Reviewers were overall enthusiastic about the findings, pending the summarized modifications below.

– Assessing the fate of RGC's daughter cells upon division is necessary to conclude on a change of neurogenic output upon loss of Diaph3. The conventional view that spindle orientation is causal to the division mode (in term of fate) of RGC daughters was recently challenged by Matzusaki, so authors should avoid strict correlated asymmetric division with neurogenic division (see page 8, line 150 and data provided in figure 3H-I). It is also important to take this into account when interpretating Diaph3 cKO in figure 4J. In this context it is quite misleading to use 'proliferative' and 'neurogenic' to qualify the division patterns based on mitotic spindle orientation. The authors should better focus on neurogenesis itself using ad hoc labeling (EdU cell cycle exit for instance) or even better, clonal analyses.

– It is unclear whether the specific loss of Diaph3 in cortical cells, which triggers apoptosis of progenitors, interferes with the assessment of the cell type composition of the cKO developing cortex (progenitors and newborn neurons).

– The study does not provide sufficient justification for the focus on SPAG5, or even why there is dysregulation of mitotic factors with Diaph3 mutated. Do DIAPH3 and SPAG5 physically interact? Why does knockdown of either protein lead to a decrease in the other? Is this at the protein and or RNA levels? This could be studied by co-immunoprecipitation for instance, both in vitro and from physiologically relevant tissues (like embryonic brain). Can this be quantified by PCR and WB? Could the reduction of SPG5 observed in Diaph3 KO results from loss of telencephalic cells (See Figure S2)? Figure 2D: Is the reduction of SPG5 mRNA mostly in the dorsal forebrain because Diaph3 expression is regionalized? The authors should provide complementary ISH to show the expression pattern of Diaph3 at the same developmental stage.

– How do to the anatomic neocortical changes relate to behavioural analyses?

[Editors' note: further revisions were suggested prior to acceptance, as described below.]

Thank you for submitting your article "DIAPH3 deficiency links microtubules to mitotic errors, defective neurogenesis, and brain dysfunction" for consideration by *eLife*. Your article has been re-evaluated by 3 peer reviewers, and the evaluation has been overseen by Joseph Gleeson as the Reviewing Editor and Jonathan Cooper as the Senior Editor. The following individual involved in review of your submission has agreed to reveal their identity: Laurent Nguyen (Reviewer #1).

Summary:

The reviewers have discussed their reviews with one another, and the Reviewing Editor has drafted this to help you prepare a revised submission. The updated manuscript includes novel experimental data that answers many of the reviewer questions. Reviewers comment that the new manuscript now includes many of the requested controls, and the BrdU experiments in Figure 4 provide stronger evidence that progenitors in mDiaph3 cKO mice undergo neurogenic divisions at the expense of symmetric proliferation. However, there were several issues that reviewers wished to point out, that could further improve the manuscript.

1. In Figure 3, there are no control IF images that accompany the shRNA knockdown of Spag5. While the western in Figure 3D shows the efficiency confirms the efficiency of shRNA knockdown, it would be useful to have the images of controls accompanying 3A-C, rather than referring the reader back to the images in Figure 1 (Lines 148-149).

2. Two separate reviewers commented that the IP in Figure 2K is not convincing. There is a faint smudge in the Diaph3 IP lane, but it looks to migrate at a different molecular weight than the Spag5 in the input lanes. If this is best example of the IP, I suggest removing it to avoid any confusion for the reader. How about IP with SPAG5, does it pull down DIAPH3? What does FT mean?

3. The cell cycle exit analysis is a welcome addition. But the authors should present data and analysis using BrdU^+^ Ki67 – / BrdU^+^ (cf Chenn and Walsh Science 2002) – cell cycle exit and not reentry.

4. Regarding their experiments on spindle pole orientation, the authors should not only remove the words 'proliferative' and 'neurogenic' to qualify the division patterns, but also asymmetric and symmetric, which are equally misleading and not addressed by the data shown.

5. Specificity of shRNA, especially SPAG5, remains problematic.

---

## [Author Response]

[…] Reviewers were overall enthusiastic about the findings, pending the summarized modifications below.– Assessing the fate of RGC's daughter cells upon division is necessary to conclude on a change of neurogenic output upon loss of Diaph3. The conventional view that spindle orientation is causal to the division mode (in term of fate) of RGC daughters was recently challenged by Matzusaki, so authors should avoid strict correlated asymmetric division with neurogenic division (see page 8, line 150 and data provided in figure 3H-I). It is also important to take this into account when interpretating Diaph3 cKO in figure 4J. In this context it is quite misleading to use 'proliferative' and 'neurogenic' to qualify the division patterns based on mitotic spindle orientation. The authors should better focus on neurogenesis itself using ad hoc labeling (EdU cell cycle exit for instance) or even better, clonal analyses.

We agree that the link between spindle orientation and fate determination of neural progenitors largely accepted in flies and initially proposed by Chenn and McConnell in the mammalian cerebral cortex has been disputed. The use of proliferative/neurogenic based on spindle orientation is not adequate and we have adapted the text accordingly. We have removed the reference to proliferative and neurogenic division in this context.

To assess neurogenesis, we have injected BrdU at e13.5, collected the brains at e14.5 and immunostained cortical sections, counted the number of BrdU^+^, Ki67^+^, and Tbr1^+^ cells, and calculated three ratios namely: (1) BrdU^+^Ki67^+^/ BrdU^+^, (2) BrdU^+^Tbr1^+^/BrdU^+^, et (3) BrdU^+^Tbr1^+^/BrdU^+^ to BrdU^+^Ki67^+^/BrdU^+^. We found that Diaph3-deficient progenitors undergo more neurogenic divisions at the expense of proliferative divisions between e.13.5 and e14.5. The results are shown in revised Figure 4O-R. The text has been amended accordingly.

– It is unclear whether the specific loss of Diaph3 in cortical cells, which triggers apoptosis of progenitors, interferes with the assessment of the cell type composition of the cKO developing cortex (progenitors and newborn neurons).

Yes. Cortex-specific deletion of Diaph3 by Emx1-Cre starts as early as e9.5 and leads to loss of neural progenitor cells. This depletes progressively the pool of progenitors, which impacts on the cell type composition of the developing mutant cortex. However, apoptosis cannot account for alterations of the spindle orientation especially if we consider that fate decisions and spindle orientation are completely unrelated.

– The study does not provide sufficient justification for the focus on SPAG5, or even why there is dysregulation of mitotic factors with Diaph3 mutated. Do DIAPH3 and SPAG5 physically interact? Why does knockdown of either protein lead to a decrease in the other? Is this at the protein and or RNA levels? This could be studied by co-immunoprecipitation for instance, both in vitro and from physiologically relevant tissues (like embryonic brain). Can this be quantified by PCR and WB?

Abnormalities in karyokinesis and disruption of (astral and spindle) microtubules in Diaph3KO/KD cells led us to test expression of key proteins involved in microtubules-kinetochore (attachment and segregation of chromosomes) and microtubules-cell cortex (orientation of spindle) interactions. Among those, GPSM2, NUMB, PAR3, SPAG5 and KNSTRN had altered expression and/or distribution. We focused on SPAG5 for the following reasons:

1. SPAG5 displayed the most arresting change in partition in *Diaph3* KO/KD cells;

2. Knock down of DIAPH3 or SPAG5 produce similar (if not identical) alterations in spindle polarity and chromosome segregation;

3. SPAG5 and DIAPH3 have the same expression pattern in the developing cortex and overlapping subcellular distribution during mitosis;

4. Contrary to GPSM2, NUMB, PAR3, nothing is known about the function of SPAG5 in the biology of neural progenitor cells.

We have now conducted Co-IP experiments, which uncover a “physical” endogenous interaction between the two proteins. This result is shown in revised Figure 2K.

Could the reduction of SPG5 observed in Diaph3 KO results from loss of telencephalic cells (See Figure S2)?

This is possible. However, i) the expression of SPAG5 (protein and mRNA) was normalized to GAPDH. Therefore, the reduction of SPAG5 is calculated relative to the amount of protein/mRNA in surviving cells only and compared between control and *Diaph3* cKO; and ii) the reduction in SPAG5 was also observed in vitro in DIAPH3-knockdown cells (Please see Figure 1 E, F and Figure 2 C-H).

Figure 2D: Is the reduction of SPG5 mRNA mostly in the dorsal forebrain because Diaph3 expression is regionalized? The authors should provide complementary ISH to show the expression pattern of Diaph3 at the same developmental stage.

All experiments of the study including ISH were conducted in conditional knockouts. The reference to KO was misleading and has been corrected throughout the text and figures.

– How do to the anatomic neocortical changes relate to behavioural analyses?

Emx1 encodes a transcription factor expressed in progenitors of the pallial subdivisions of the telencephalon. Linage tracing using Emx1-Cre mouse line showed that Emx1 derivatives are glutamatergic neurons of cortical areas such as the neocortex, hippocampus, pallial amygdala, and entorhinal cortex among others (Gorski et al., 2002; Cocas et al., 2009). Interestingly, regions that play important role in social interactions are the prefrontal cortex (PFC) and the basolateral amygdala (BLA) (Huang et al., 2016 and 2020). Patients suffering ASD have alterations in PFC activity and/or connectivity and sensorimotor dysfunctions causing social and motor deficits respectively (Gillbert et al., 2008; Ha et al., 2015; Mosconi and Sweeney 2015; Barak and Feng 2016; Bicks et al., 2015). These regions are interconnected, and their glutamatergic neurons derive from Emx1 progenitors (Gorski et al., 2002; Cocas et al., 2009). In this current work, we showed that Diaph3 is necessary for the normal development of all cortical areas. *Diaph3* conditional knockout mice have severe anatomic changes in cortical areas implicated in social interactions and motor control. Notably, they have reduced number of cortical excitatory neurons. This can affect the excitatory/inhibitory balance and explain the lower sociability and locomotor activity in these mice.

[Editors' note: further revisions were suggested prior to acceptance, as described below.]

Summary:The reviewers have discussed their reviews with one another, and the Reviewing Editor has drafted this to help you prepare a revised submission. The updated manuscript includes novel experimental data that answers many of the reviewer questions. Reviewers comment that the new manuscript now includes many of the requested controls, and the BrdU experiments in Figure 4 provide stronger evidence that progenitors in mDiaph3 cKO mice undergo neurogenic divisions at the expense of symmetric proliferation. However, there were several issues that reviewers wished to point out, that could further improve the manuscript.1. In Figure 3, there are no control IF images that accompany the shRNA knockdown of Spag5. While the western in Figure 3D shows the efficiency confirms the efficiency of shRNA knockdown, it would be useful to have the images of controls accompanying 3A-C, rather than referring the reader back to the images in Figure 1 (Lines 148-149).

Control images for shRNA knockdown were included in revised Figure 3 panel A.

2. Two separate reviewers commented that the IP in Figure 2K is not convincing. There is a faint smudge in the Diaph3 IP lane, but it looks to migrate at a different molecular weight than the Spag5 in the input lanes. If this is best example of the IP, I suggest removing it to avoid any confusion for the reader. How about IP with SPAG5, does it pull down DIAPH3? What does FT mean?

We agree. It is not the most convincing result of the paper. Co-IP experiments depend heavily on the quality of antibodies and an excellent antibody for WB or IF is not necessarily suitable for co-IP, especially for endogenous proteins. We have therefore removed this data altogether. We believe that the rest of data stand on its own without this. The results of WB, RT-qPCR, ISH and IF are extremely strong. Furthermore, the function of SPAG5 has never been explored in the brain. All these reasons justified the choice of SPAG5 for further investigations in the paper.

3. The cell cycle exit analysis is a welcome addition. But the authors should present data and analysis using BrdU^+^ Ki67 – / BrdU^+^ (cf Chenn and Walsh Science 2002) – cell cycle exit and not reentry.

Thank you for pointing this out. We have now included this data in the revised Figure 4 Panels Q and R and the text has been amended accordingly. Our results show that the percentage of precursors that exit cell cycle (BrdU^+^Ki67^-^/BrdU^+^) is enhanced in *Diaph3* cKO compared with controls (+20%, Figure 4Q). Also, the ratio of non-cycling to cycling cells (BrdU^+^Ki67^-^/ BrdU^+^Ki67^+^) is *increased in Diaph3* cKO compared with controls (+40%, Figure 4R).

4. Regarding their experiments on spindle pole orientation, the authors should not only remove the words 'proliferative' and 'neurogenic' to qualify the division patterns, but also asymmetric and symmetric, which are equally misleading and not addressed by the data shown.

We have replaced symmetric by planar (mitotic spindle parallel to the ventricular surface) and asymmetric by non-planar (spindle orthogonal or oblique to ventricular surface) in Figure 4 and throughout the text.

5. Specificity of shRNA, especially SPAG5, remains problematic.

We regret that we fail to convince the reviewers of the specificity of SPAG5 shRNA despite the use of scrambled shRNA and beta-catenin as internal control. Expression of beta-catenin is diminished in SPAG5 knock-down cells (Liu et al., 2019) and also in our results (Figure 3D).